# Microbial Etiology and Prevention of Dental Caries: Exploiting Natural Products to Inhibit Cariogenic Biofilms

**DOI:** 10.3390/pathogens9070569

**Published:** 2020-07-14

**Authors:** Xiuqin Chen, Eric Banan-Mwine Daliri, Namhyeon Kim, Jong-Rae Kim, Daesang Yoo, Deog-Hwan Oh

**Affiliations:** 1Department of Food Science and Biotechnology, College of Agriculture and Life Sciences, Kangwon National University, Chuncheon 200-701, Korea; chenxiuqin0127@kangwon.ac.kr (X.C.); ericdaliri@kangwon.ac.kr (E.B.-M.D.); namhyeon@kangwon.ac.kr (N.K.); 2Hanmi Natural Nutrition Co., LTD 44-20, Tongil-ro 1888 beon-gil, Munsan, Paju, Gyeonggi 10808, Korea; chief1111@daum.net; 3H-FOOD, 108-66, 390 gil, Jingun Oh Nam-Ro, Nam Yang, Ju-Shi, Gyung Gi-Do 12041, Korea; daesangy@naver.com

**Keywords:** dental caries, biofilm, cariogenic microorganisms, antimicrobial compounds

## Abstract

Dental caries is one of the most common microbe-mediated oral diseases in human beings. At present, the accepted etiology of caries is based on a four-factor theory that includes oral microorganisms, oral environment, host, and time. Excessive exposure to dietary carbohydrates leads to the accumulation of acid-producing and acid-resistant microorganisms in the mouth. Dental caries is driven by dysbiosis of the dental biofilm adherent to the enamel surface. Effective preventive methods include inhibiting the cariogenic microorganisms, treatment with an anti-biofilm agent, and sugar intake control. The goal is to reduce the total amount of biofilm or the levels of specific pathogens. Natural products could be recommended for preventing dental caries, since they may possess fewer side effects in comparison with synthetic antimicrobials. Herein, the mechanisms of oral microbial community development and functional specialization are discussed. We highlight the application of widely explored natural products in the last five years for their ability to inhibit cariogenic microorganisms.

## 1. Introduction

The human body is home to trillions of microbes, and the oral cavity is one of the largest sources of microbes. There are around 700 to 1000 microbial species that colonize the human mouth. The occurrence and development of oral diseases such as dental caries, periodontal disease, and oral cancer are closely related to oral microorganisms [1,2,3,4]. Meanwhile, oral microorganisms can enter the blood circulatory system through a damaged oral mucosa and cause a rise of systemic antibody levels thereby increasing the risk of a variety of cardiovascular diseases [2]. Peres et al. evaluated the societal relevance of preventing and addressing oral diseases worldwide; the direct costs of dental diseases were estimated at $356.80 billion and the indirect costs at $187.61 billion in 2015 alone [3]. The three diseases with the highest direct and indirect costs of prevention and treatment worldwide in 2015 could be ranked as follows: diabetes (€119 billion) > cardiovascular diseases (€111 billion) > dental diseases (€90) billion. 

Dental disease is undoubtedly a public health problem and is among the most prevalent diseases globally, in particular, dental caries which is a biofilm-associated disease [5]. The World Health Organization reported that 60 to 90%of school children and nearly 100% of adults worldwide are suffering from cavities [6]. Therefore, the prevention of caries plays an important role in public health management. Federation Dentaire Internationale (FDI) presented the minimal intervention dentistry definition on managing dental caries in 2002, emphasizing that the existing preventive measure is to maintain a healthy tooth structure as much as possible [7]. With the recent recommendation for early detection and monitoring of caries, rather than waiting until a cavity is formed, the prevention of caries was shifted from the surgical model to a medical model, and the proportion of individuals receiving preventive oral health care has been increasing in recent years [8]. Preventing caries preserves a sound tooth structure, prevents the demineralization of enamel, and promotes natural healing processes [9]. Caries risk assessment contributes to determining the specific protective factors against dental caries and the need for therapeutic intervention. John Featherstone et al. pointed out that disease indicators such as ‘bad bacteria’, absence of saliva, and poor dietary habits are determinants of caries. They demonstrated that saliva, sealants, antibacterials, fluoride, and a controlled diet can contribute to keeping the teeth healthy, and each of the above strategies alone can be used to prevent dental caries [10]. The Alliance for a Cavity-Free Future has proposed steps to prevent dental caries, which include balancing the levels of oral bacteria, controlling the consumption of sugary and starchy foods, strengthening the demineralized enamel though fluoridated products [11]. Improvement of the oral flora is one of the effective strategies to prevent dental caries. In this study, we discuss the prevention of caries through the inhibition of dental biofilms. The oral biofilm formed by cariogenic (that is, pathogenic) microorganisms is a complex microbial community in the mouth. Numerous studies have reported that the difference between pathogenic biofilms and non-pathogenic biofilms corresponds to the proportion of cariogenic microorganisms [12]. The accumulation of pathogenic biofilm is one of the main causes of dental caries. Therefore, agents with anti-biofilm properties have been proven to be effective in preventing dental caries [13]. We here describe strategies used for controlling cariogenic microorganisms.

### 1.1. Chemical Agents

In recent years, many chemical agents have been reported to have effects on the metabolism of bacteria and the adherence of bacterial cells. Some, such as chlorhexidine [14,15,16,17], delmopinol [17,18], and triclosan [19], have shown potent inhibitory activities against the development and maturation of biofilm. It is generally believed that the mechanism of chlorhexidine bactericidal activity is the destruction of the serosa permeability barrier against bacterial cells. Low concentrations of this agent can lead to partial cytoplasmic leakage, while high concentrations cause cytoplasm condensation and denaturation and, thus, sterilization. Fluorides such as amine fluoride [20,21], sodium fluoride [21], and stannous fluoride [22] act as powerful agents in the prevention of dental caries. Free fluoride ions in sodium fluoride may interfere with bacterial metabolism through bacterial cell membranes [23]. Amine fluoride is a cationic antimicrobial compound with unclear mode of action [22]. It is suggested that amine fluorides bind to the surfaces of bacterial cells and disturb the stability of the bacterial membrane [20]. 

However, chemical agents may cause considerable side effects [24,25]. The limited penetration of antimicrobial agents may restrict their inhibitory effects on cariogenic microorganisms. Also, the concentration of agents and the duration of exposure affect their anti-caries efficiency [26]. A high concentration of chemical agents can unbalance the oral flora and have adverse side effects such as vomiting, diarrhea, mucosal desquamation [27], and tooth staining [28]. Therefore, exploiting alternative natural products as preventive measures for dental caries could be promising for dental caring.

### 1.2. Natural Products, Plant Extracts, and Probiotics

Clinical trials have proved the effectiveness of several natural compounds in the treatment of dental caries, among which catechol, emetine, quinine, and flavone are the most reported [29,30]. Phytochemicals isolated from plants that can be used as effective and economical treatments are needed [31]. However, many plant products such as herbs and spices show toxicity to cells, and hence, cytotoxicity tests and dose controls are needed for their safe use [32]. According to the randomized controlled clinical trial of Usha et al. (2017), a 0.5% extract of *Stevia rebaudiana* leaves reduced cariogenic organisms significantly and promoted the buffering capacity of the saliva in patients at high for caries [33]. In recent years, researches on probiotics beneficial for the buccal cavity have been increasing. It is reported that certain probiotics can inhibit cariogenic microorganisms by the production of microcin [34], hydrogen peroxide [35], and bacteriocins [36]. Rodríguez et al. (2016) compared milk supplemented with probiotic lactobacilli with standard milk for their effects of high patients at risk for caries. The results showed that group receiving the probiotic had less new lesions than the standard milk group, demonstrating the possible use of probiotic lactobacilli for caries control [37]. This review aims to present the characteristics of the cariogenic biofilm, as well as to summarize systematically the application of probiotics and plants including herbs and spices for the prevention of caries in the past five years and explain their mechanism of action, as a reference for further research in the area of dental caries prevention.

## 2. Supragingival Microbial Biofilms and Dental Caries

### 2.1. Oral Microbiota 

In 1890, Miller first proposed the “chemical bacteria theory” for the occurrence of dental caries in the book titled “Microbes in the human mouth” [38], suggesting that the dental biofilm is composed of microorganisms. Oral microorganisms, including bacteria, yeasts, viruses, mycoplasmas, protozoa, and archaea form a heterogeneous ecological system in the mouth, which is known as oral microbiota [39,40]. The oral cavity presents a warm and nutritious environment for the oral microbiota and, at the same time, it controls bacterial colonization to avoid invading pathogenic microbes. The oral microbiota plays a critical role in maintaining oral health [41]. However, under certain conditions, invading microorganisms cause an imbalance of the host’s commensal microbial community, which results in dental disease. 

### 2.2. Dental Biofilms 

The oral microbiota on teeth surface tends to form polymicrobial communities, known as dental biofilm [42]. It is now clear that the matrix of extracellular polymers (EPS) provides a pathological habitat for cariogenic microorganisms. A large body of evidence indicates that dental caries is essentially a biofilm-induced disease, rather than an infectious disease [43], and the disease process begins in the biofilm that covers the surface of the tooth [5]. Caries biofilm (biofilm that may cause caries) is an extremely active and complicated ecosystem, rich in EPS (Figure 1). The formation of the biofilm begins when a salivary glycoprotein film (called dental pellicle) coats a tooth surface [20]. Gram-positive bacteria including streptococci of the *mitis* [44] and *mutans* species [45] (which are considered as initial colonizers of the biofilm) then form EPS, which enhance the adherence of other organisms. Emerging evidence shows that acid-producing bacterial species of the genera *Veillonella* [46], *Scardovia* [47], *Lactobacillus* [48], and *Propionibacterium* [49] could be present in the dental biofilm as colonizers and may induce in cariogenic conditions in the mouth. 

The EPS provide new binding sites for other acid-producing microorganisms and enhance their virulence [44]. Early studies focused on the microbial composition of cariogenic biofilms, but it is now being increasingly recognized that the structural and biochemical properties of EPS play important roles in the etiology of caries [50]. The matrix of EPS provides protection and mechanical stability, making the biofilm recalcitrant to antimicrobials and difficult to remove. The microbes are embedded in a substrate of EPS and constantly produce acids that are physically protected from the rapid buffering of saliva [51]. The studies on caries have focused on microbial behavior in biofilm communities using experimental biofilm models that can simulate the metabolic processes during carbohydrate exposure in the mouth and assessing the dose–response sensitivity of anti-caries agents [43]. In other words, this strategy can help to investigate the cariogenicity of dietary sugars and to evaluate the anti-caries effects of substances in vitro [52].

*Streptococcus mutans* biofilm has been largely accepted to have cariogenic potential [53,54] which depends on three core attributes: (i) acid production, (ii) acid resistance—which makes it able not only to metabolize a wide range of carbohydrates into organic acids but also to thrive under low pH conditions [55]—and (iii) the ability to synthesize EPS, which can be seen as a growth-promoting process, providing protection for cells and thus allowing them to survive in harsh environments [56]. Three glucosyltransferases (*Gtf*BCD) are matrix-producing enzymes of *S. mutans* that are involved in the establishment of a cariogenic biofilm [57]. However, as the science of prevention and treatment of dental caries has evolved, it has become clear that simply targeting *S. mutans* and limiting sugar intake is not sufficient to prevent caries. The major EPS components in cariogenic biofilms are polysaccharides, particularly *S. mutans*-derived glucans as well as soluble glucans and fructans produced by other species (e.g., *Actinomyces*, *Streptococcus salivarius*, and *Streptococcus gordonii*) [47,50]. Recent molecular analyses have revealed the presence of a pathogenic flora that includes bacteria different from streptococci (e.g., *Scardovia* and *Actinomyces*) and fungi (e.g., *Candida albicans*) [58,59]. In addition to *S. mutans*, *Lactobacillus* [48], *Bifidobacterium* [60], and *Scardovia* species [47] are also considered as caries-associated colonizers. Data from previous studies have suggested that the susceptibility of biofilms to antibiotics, preservatives, or anti-adhesion compounds is closely related to microbial diversity [61,62,63]. Due to the strong competitiveness of cariogenic microorganisms, microbial abundance decreases during the maturation of cariogenic biofilms [50]. The etiology of caries is attributed to the dominance of cariogenic microorganisms over health-associated commensal species. Thus, challenges for the prevention of dental caries are posed by the complexity of the biofilm matrix as well as the abundance of microorganisms.

### 2.3. Microbial Etiology of Dental Caries 

At present, it is accepted that dental caries result from a complex interaction between acid-producing microorganisms and fermentable carbohydrates over time [40]. Although the oral microbiome influences the formation of dental caries, many host factors including teeth and saliva also affect caries development, leading to a disease that tends to be chronic and slowly progressive (Figure 2.). The dental biofilm is an important component in the etiology of dental caries. The complexity of the matrix, the transfer of resistance genes, as well as the physical protection provided by EPS are risk factors for caries. Numerous studies have reported that controlling the dental biofilm is the key to preventing tooth decay [64]; additional challenges are posed by the lack of a single obvious target for therapeutic intervention and by the poor retention of locally administered treatments [44].

## 3. Recent Advances in Natural Antimicrobial Compounds for the Prevention of Dental Caries

### 3.1. Plant-Derived Cariogenic Biofilm Inhibitors

There has been a growing interest in plants that are rich in natural antimicrobial compounds [65]. Folk dental practitioners have realized the importance of medicinal plants as effective providers of drugs. Some plants lacking unwanted side effects have even shown higher efficiency than synthetic drugs in the inhibition of dental caries [66]. The World Health Organization has reported that about 80% of the world’s population rely on herbal products to treat some diseases [67], and most of the herbal medicines contain at least one botanical molecule [68]. It is worth mentioning that the bioactivity, and bioavailability of phytochemicals have been explored widely [53,66]. According to the research of Malvania et al. (2019), a licorice extract produced a significantly higher inhibitory effect on oral pathogens when compared to sodium fluoride [66]. Fluid or dry plant extracts are added to oral caring products, such as toothpaste, mouthwash, and oral care functional food, to enhance their anti-caries properties [69]. Meanwhile, plant ingredients are also used while filling cavities to treat caries pain [70]. Most of the antibacterial substances in plants are secondary metabolites that are not necessary for plant growth but have special physiological functions. They usually include alkaloids, phenols, flavonoids, and organic acids [71]. Research on the mechanisms of the anti-caries action of plant-derived compounds, as well as on their biological effects on the host, is still required. Over the years, plants have been used as natural therapies beneficial to oral health, some of them having antibacterial properties and reducing infections [72]. Another action of anti-caries compounds is the inhibition of glucosyltransferase (which plays a key role in the synthesis of water-insoluble glucan) to prevent the formation of cariogenic biofilms [73,74]. It is necessary to investigate plant extracts, which contain many different bioactive compounds. Researches have indicated that cinnamon, clove, and coriander are effective in preventing dental caries [75]. The antimicrobial activity of phenolic acids is related to the number and position of substituents on the benzene ring. The saturation and length of the side chains may influence the antimicrobial potential against oral pathogens [76]. One of the probable antibacterial mechanisms of xanthorrhizol was found to be the formation of hydrogen bonds between the hydroxyl groups in xanthorrhizol and proteins in the cell membrane. The hydroxyl groups bind to the cell membrane of *C. albicans*, affecting its membrane permeability and eventually leading to cell lysis [77]. It is, however, known that one of the mechanisms of the antimicrobial action of alkaloids is the suppression of the pathogen cytokinesis. Polyphenols are known to play a role in the inactivation of cellular enzymes in pathogens [78,79]. More studies are, however, required to unveil more details about the mechanism of action of anti-caries bioactive compounds. In addition to their antibacterial activity, plants often possess natural antioxidants due to the presence of polyphenols and flavonoids [32]. Natural plants can be an adjunct therapy to mechanical dental biofilm control. Whole or specific parts of various plants have been used in the prevention of dental biofilm formation and could reduce the high global incidence of dental caries. Some trials regarding plants, listed in Table 1, have achieved favorable outcomes for dental caries prevention.

#### 3.1.1. Effect on Bacterial Growth

Bioactive compounds from plants have been examined for their ability to inhibit the growth of cariogenic microorganisms [78,87,88,89,90]. An essential oil has been demonstrated to have pharmacological properties such as antibacterial potential [91]. Many plant extracts with complex chemical composition, including alkaloids, avonoids, isoavonoids, tannins, cumarins, glycosides, terpens, phenolic, phenylpropanol, monoterpenaldehyde, and monoterpene alcohol, can be used for caries prevention purposes [92]. Bodiba et al. (2018) demonstrated the possible future use of the extract of *Pongamia pinnata*, *Azadirachta indica*, *Psidium guajava*, and *Mangifera indica* for the prevention of dental caries. These researchers tested the antibacterial activity of the herbs mentioned above against *S. mutans* using a microdilution method. Using the checkerboard method to measure the synergistic ability of the herbs, they proved the important role of herbs in dentistry [32]. Phytochemicals isolated from certain herbs used in traditional medicine have been proposed as potential alternatives against cariogenic microbes that cause dental caries [90]. Antimicrobial agents including lipophilic alkylamides present in various herbs have been studied in clinical trials [93]. Also, other antimicrobial compounds in herbal medicines have been shown to display antimicrobial activity against oral pathogens, as well as prevent the release of histamine and the accumulation of cariogenic microorganisms on teeth surface.

#### 3.1.2. Alteration of Initial Adhesion, Aggregation, and Integrity

The first step against biofilm formation is the use of biosurfactants and bioemulsifiers to change the physical and chemical properties of the cell surface, thereby weakening the adhesion of microorganisms. Glycyrrhizin inhibits the adherence of *S. mutans* by affecting the activity of glucosyltransferase. The mechanisms of the antimicrobial and anti-biofilm actions of glycosides can involve the inhibition of Sortase A and Sortase A-mediated aggregation of *S. mutans* [84,85]. Padma hepaten is a traditional Tibetan medicine that contains an efficient polyphenolic formula derived from several herbs. Padma hepaten bioactive compounds act by reducing the cariogenic biofilm via the downregulation of the genes (gtfB, gtfC, and ftf) that code for EPS [74]. Janakiram et al. (2020) compared herbal toothpastes with non-herbal toothpastes by searching databases of randomized controlled trials and found that herbal toothpastes were superior to the non-herbal toothpastes in dental biofilm reduction [94].

#### 3.1.3. Modulation of Bacterial Quorum Sensing

Quorum sensing (QS) is a key regulator of virulence in cariogenic biofilms. Biofilm formation is based on the signal-mediated QS system. Plant extracts can inhibit QS genes and QS-controlled factors and interfere with biofilm accumulation. They can also target several pathways of bacterial metabolism. Choi, H. et al. (2017) found that *Camellia japonica* and *Thuja orientalis* methanolic extracts have potential anti-quorum-sensing abilities against oral pathogens [78]. When Philip, Nebu, et al. (2019) investigated the effects of cranberry extracts on the virulence of *S. mutans*–*C. albicans* biofilms, the results showed that polyphenol-rich cranberry extracts significantly reduced the acidogenicity and metabolic activity of these biofilms [59]. 

### 3.2. Microbial Cariogenic Biofilm Inhibitors—Probiotics

Since the solubilization of tooth minerals by microbial metabolic acids is irreversible, and treatment without prevention is not sustainable [95], there has been increasing interest in the possible effects of probiotics on the prevention of dental caries in the past few years [96]. The development and evaluation of probiotics and probiotics containing oral symbiosis products will be an important topic for the management of dental caries in the future [97,98].

Probiotics are live microorganisms that are beneficial to the host by colonizing the human body when administered in adequate amount [99]. Probiotics can change the composition of the microbial communities in a certain organ or tissue of the host [100]. Homeostasis and dysbiosis of oral microbial communities ultimately lead to health or disease, respectively [62]. However, it is difficult for exogenous probiotic bacteria to colonize in the established oral microbiota [101]. Therefore, overcoming the limitations to probiotic colonization in the oral cavity is a challenge. Bacteria naturally present in the mouth can show dual probiotic effects, inhibit the growth of cariogenic species, as well as modify the pH of the oral environment [102,103,104]. The most widely researched probiotics, i.e., *Lactobacillus rhamnosus*, *Lactobacillus casei*, *Lactobacillus reuteri*, *Lactobacillus plantarum*, *Lactobacillus brevis*, *Bifidobacterium lactis*, have been investigated, and their capacity to reduce cariogenic pathogen count and control plaque pH has been assessed [98]. In recent years, *L. rhamnosus*, *L. reuteri*, and *B. lactis* have been examined in clinical trials after incorporation in dental caring products, such as tablets, lozenges, and chewing gums. Their mechanism of anti-caries action is competition for essential nutrients [97]. Meanwhile, the production of antibacterial factors including bacteriocin, organic acids, and hydrogen peroxide can protect the host from the overgrowth of pathogens [105]. Bacteriocins such as nisin, pediocin, and reuterine are known bioactive compounds produced by various probiotic strains, which are effective against oral pathogens [106]. Recent findings related to probiotic achieving favorable outcomes in dental caries prevention are listed in Table 2

### 3.3. Incorporation of Natural Antimicrobials in Caries 

The incorporation of natural products in treatments for the prevention of caries might reduce the therapeutic costs, while causing minimal side effects. However, in vivo toxicity studies and clinical trials are still necessary. The studies carried out in recent decades have confirmed the anti-caries role of probiotics and natural compounds extracted from herbs and spices [75]. Their main action is based in three effects: reduce bacterial growth rate, reduce the adhesion ability of pathogens, and inhibit the enzymatic activity of glucosyltransferase and amylase [72,73,88]. The treatment with a single natural product dose—dependently inhibited the cariogenic biofilm, while plant extracts in combination with probiotics demonstrated synergistic effects. Ping et al. (2008) reported that the combination of green tea extract and probiotics produced a more significant pathogen reduction than probiotics or plant extracts used separately [111]. Wang et al. (2019) demonstrated that the cooperative fermentation of probiotics and a Chinese herbal medicine had a synergistic antifungal effect [83]. A synergistic antibacterial effect was observed by combining *Azadirachta indica*, *Pongamia pinnata*, *Psidium guajava*, and *Mangifera indica* against *S. mutans* [32]. In another study, a combination of gingerol and allicin produced a great antimicrobial action [83]. To control both effectiveness and safety, the relationship between the oral environment and the anti-microbial activity of bioactive compounds, as well as the synergistic/antagonistic effects of natural antimicrobials, still need to be explored in details.

## 4. Conclusion and Future Perspective

The present article reveals that natural antimicrobial agents such as probiotics, herbs, and spices appear to be effective in controlling dental caries. Dental caring products containing extracts of natural plants have been on the market for many years. Also, probiotics not only serve as potential antimicrobial agents but also maintain the stability of the oral ecosystem. This review suggests that individual treatments using single herbal products or probiotics act on distinct targets; therefore, it may be more effective to combine several plants or to combine plants with probiotics. The development of functional products combining probiotics and polyphenol extracts could be an interesting research direction in the food industry [112]. Herein, we summarized the application of plant-derived natural products and microbial products for caries prevention in the recent five years and discussed the antimicrobial properties of natural products. In the future, the synergy between natural plants and microbial products must be targeted to help define novel, effective, and safe anti-caries strategies. However, efficient clinical studies are necessary for the discovery of their biofilm-interfering or -inhibiting activities.

## Figures and Tables

**Figure 1 pathogens-09-00569-f001:**
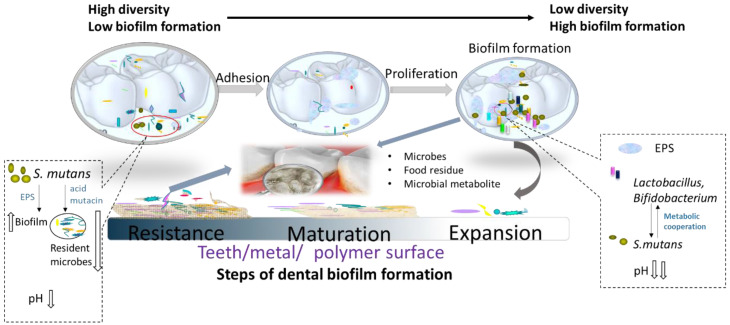
Diagram of the dental biofilm formation. EPS, extracellular polymers. *S. mutans*, *streptococcus mutans.*

**Figure 2 pathogens-09-00569-f002:**
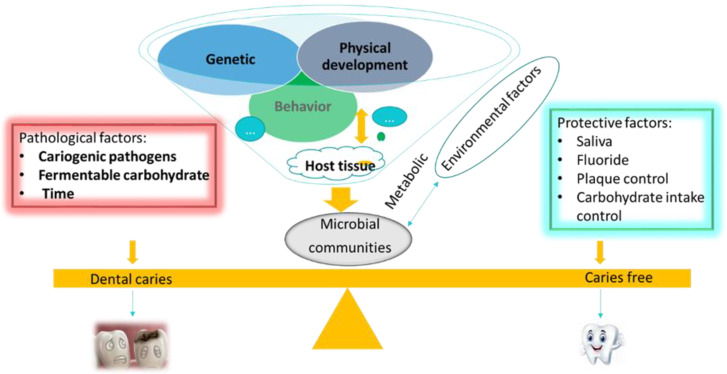
Diagram of the caries process in relation to diet.

**Table 1 pathogens-09-00569-t001:** Plant compounds/extracts and their bioactivity against cariogenic microorganisms.

No.	Plants	Extracts & Bioactive Compound	Target Organisms	Biological Activity	Reference
1	*Acacia arabica*	Ethanol, acetone, and water extract	Strong biofilm-forming strains isolated from patients	Anti-biofilm, antimicrobial	[80] 2017
2	*Tamarix aphylla L.*				
3	*Melia azedarach L.*				
4	*Bauhinia forficata*	Tincture	*Streptococcus* spp. Salivary samples from healthy volunteers	Anti-biofilm, antimicrobial	[16] 2019
5	*Bauhinia forficata*	Phenolic acids, chlorogenic acids	*S. mutans* (ATCC 25175) *Streptococcus sanguinis* (ATCC 10556) *Candida albicans* (ATCC 22972) *Fusubacterium nucleatum* (ATCC 25586) *Lactobacillus casei* (ATCC 393) *Prevotella nigrescens* (ATCC 33563), *Bifidobacterium dentium* (ATCC 27534)	Antimicrobial, anti-demineralizing	[76] 2020
6	*Curcuma xanthrorrhiza*	Ethanol extract, xanthorrhizol	*C. albicans*	Anti-biofilm, antimicrobial	[77] 2019
7.	*Cymbopogon citratus*	Lemon Grass Essential Oil	*Streptococcus agalactiae*, *Staphylococcus epidermidisand*, *Lactobacillus fermentum*	Antimicrobial, Anti-biofilm	[81] 2019
8	*Pongamia pinnata*	Methanolic extract	*S. mutans* MTCC 497, *S. mutans* MTCC 890	Antimicrobial	[82] 2017
9	*Acacia catechu*	Methanolic extract			
10	*Clove*	Eugenol, oleic acid, lipids	Microorganisms collected from extracted teeth	Antimicrobial	[83] 2016
11	*Ginger-garlic paste*	Gingerol, allicin			
12	*Tea tree*	Catechins			
13	*Camellia japonica*	Phenolic compound, flavonoid	*S. mutans* ATCC 25175 *Candida albicans* NUM961	Antimicrobial, anti-biofilm, anti-GTase	[78] 2017
14	*Thuja orientalis*				
15	*Quercus infecteria*	Tannins, cardiac glycosides, sterioids, terpenoids, alkaloids	*Lactobacillus casei*	Antimicrobial	[84] 2020
16	*Sterculia lychnophora Hance*	Organic acids, glycosides,	*S. mutans* ATCC 25175	Antimicrobial, cariogenic properties inhibition	[85] 2016
17	*Cinnamon bark*	Methanol extract, cinnamaldehyde	*C* *andida* *albicans ATCC 2091*	Antimicrobial‘	[86] 2019
18	*Cinnamomum burmannii*	Water extract	*S. mutans UA159*	Antimicrobial, anti-biofilm	[53] 2020
19	*Licorice Root*	Glycyrrhizin	*S. mutans* ATCC 25175	Antimicrobial	[66] 2019
20	*Eurycoma longifolia jack*	Ethanol extract, canthin-6-one alkaloids, β-carboline alkaloids, quassinoids	*Candida albicans*, *S. mutans*, *Lactobacillus casei*	Antifungal, Antimicrobial,	[79] 2019

**Table 2 pathogens-09-00569-t002:** Probiotics and their bioactivity against cariogenic microorganisms, examined using the biofilm model.

Probiotics	Target Bacteria	Type of Biofilm Model	Bioactive Compound/Action Mechanism	References (Year)
*L. rhamnosus SD11*	*S. mutans* *lactobacilli*	Human oral cavity	Integrate into the bacterial communities of the dental biofilm	[97] 2019
*L. salivarius*	*S. mutans* *C. albicans*	Double species, static	Strong competitor of oral pathogens	[107] 2017
*Streptococcus salivarius strain M18*	*S. mutans*	plaque-disclosing solution	Bacterins	[108] 2013
*L. casei ATCC 393*, *L. reuteri ATCC 23272*, *L. plantarum ATCC 14917*, *L. salivarius ATCC 11741*	*S. mutans* *ATCC 25175*	Single specie biofilm, dual-*S. mutans*–*Lactobacillus* spp. biofilm, static	Organic acid, peroxide	[96] 2018
*L. casei Shirota*, *L. casei LC01*, *L. plantarum ST-III* *L. paracasei LPC37*	*S. mutans* *Streptococcus spp.*, *S. sanguinis*	Multi-species biofilm, static	Alteration of the oral microbiota	[109] 2017
*L. casei 01*	*S. mutans*, *S. parasanguinis*, *S. salivarius*	Multi-species biofilm, static	Adhere to dental surfaces and integrate into the bacterial communities of the dental biofilm	[110] 2019
*Lactobacillus* *plantarum FB-T9*	*S. mutans*	Rat oral cavity	FB-T9 is a strong competitor of *S. mutans* for temporal and spatial niches	[54] 2020

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
