# Peer review of "Microbial Etiology and Prevention of Dental Caries: Exploiting Natural Products to Inhibit Cariogenic Biofilms"

_pathogens, 2020, doi:10.3390/pathogens9070569_

Round 1
Reviewer 1 Report
This is a concise review describing recent investigations using natural compounds in the management of cariogenic biofilms. Since there is increasing evidence of some adverse influences of chemical compounds such as a chlorhexidine for the control of oral biofilm, the natural compound is getting a lot of attention as an alternative ingredient. While this review might be interesting for the readers of pathogens, major modification is needed to enhance the quality of this article because of the following major reasons.
Major comments
- Title- The main title is “Prevention of dental caries”. And the subtitle is Exploiting Natural Antimicrobial Compounds as Novel Antiseptic Agents to Inhibit Cariogenic Biofilms. It would be magnificent if it was a review article on the prevention of the caries disease.
Considering the characteristics of the review article, the current concept of dental caries prevention should be introduced firstly. In general, it is necessary to outline the concept of dental caries prevention or the concept of onset / progression in the Introduction section. Because it is the reason that the main title is “Prevention of dental caries”.
Clearly, the current strategy for dental caries is based on the MI or MID proposed by FDI (2002 or 2016). Alternatively, the current development and avoidance of caries can be outlined in the Caries Balance / Imbalance Model published by Dr. John Featherstone et al. (CDAJ 35(10), 2007). In that situation, the authors should suggest to the reader what to review.
Readers will be confused if you close up only one particular factor.
Based on these, I think that the new concepts and strategies that should be updated regarding the prevention of dental caries are scarce.
In the MI concept, improvement of oral flora is only one of them (1/5 or 1/6).
You should also review whether caries can be prevented with only one of them. Or you should comment.
- If the authors would like to focus to natural antimicrobial compounds, the reviewer recommends that the sections from “2.1. Oral microbiota” to “2.3. Microbial etiology of dental caries” are minimized or included in the Introduction section. Because the mechanisms of oral microbial community and its function are well-known and already stated in many other reviews. The authors stated the antimicrobial effects of herb and spices separately in P6. The reviewer suggests that the effects of natural compounds are described according to the extracts, such as root extract, herb (polyphenols), essential oil, spice and fruit (nut).
Alternatively, how about describing in terms of antibiofilm actions such as the inhibition of adherence, bacterial growth, acid production, aggregation, interference with bacterial metabolism (glucosyltransferases) and virulence factor (quorum sensing)?
There seems to be a problem with the integrity of the content and title.
Would you like to refer to the mini-review of Cited Reference No. 36(Baker and Edlund)?
It is possible to consider that the disease name of dental caries is not expressed.
Minor comments
- P3, L99-100, L105: Generally, the term tooth caries is not used. Dental caries is common. (Dental Caries 2nd ed, Blackwell Munksgaard: 2008), Generally, the term pioneer colonizers is not used. Initial colonizer or early colonizer is common. (Kolenbrander PE and London J. J Bacteriology 175(11), 1993)
- Table 1 shows the results of in vitro or ex vivo Some randomized clinical trials of natural products are reported in recent five years. Please create a summary of the studies with a separate sheet. If the ingredients include in a commercially available product, the product name should be provided. The readers cannot understand that some natural compounds may have oral health benefits, unless an overview of the strength of clinical evidence is provided.
- Please provide the adverse effects and events. Especially, cytotoxity compared with the bactericides is important. For example, the test was performed at least in studies of reference #14, 76 and 85 cited by the authors.
- The reviewer think that some inappropriate papers were cited. Please provide the search strategy and selection criteria for this review.
- P7-3.2. Microbial sources-Probiotics
No description was appeared in this section.
- Section 1.1. Vaccines
Although DNA vaccines are interesting, this section is not needed because of no relationship with the topics of natural compounds.
- The authors need to confirm the article carefully again. The overstatements and wrong citations were found at several sentences.
The examples are as follows:
P6, L209- Spice essential oil has been… antifungal potential [82].
This sentence was overstated. The authors tested only the anti-bacterial activity of three essential oils using a disc diffusion test. They did not demonstrate any anti-inflammatory and antifungal activity. And citation was wrong.
Ref. #82
John Rozar Raj, B.; Geetha,R.V.; Lakshmi, T. Anti-bacterial activity of three essential oils- An in vitro study. Int. J. Res. Pharm. Sci. 2019, 10, 1049-1053.
P6, L212-Ref. #83
The authors tested antimicrobial activity against non-oral microorganisms of only three essential oils such as a clove, cinnamon and pepper.
P7. L222-Ref. #64
Malvania et al. have determined the MIC and zone of inhibition of licorice root extract against S. mutans. They did not analyze the activity of glucosyltransferase.
Ref. #84 Journal name is wrong.
Author Response
Response to the Reviewers’ comments
We are very thankful for your time and valuable suggestions on the manuscript. We appreciate the important comments and we are glad to respond to the comments highlighted by you. Corrections and detailed information have been highlighted in red in the revised manuscript. The response about the comments as follows:
Response for Major comments:
1.
We agree that the new concepts and strategies regarding the prevention of dental caries are necessary for our manuscript. Therefore, we have added more information about the current concept of dental caries prevention in the revised manuscript. Besides, we carefully reviewed the valuable reference published by Dr. John Featherstone et al., the authors agree that caries risk assessment is an important part for therapeutic intervention, the article is worth recommending to readers, the update regarding the prevention of dental caries are added. In our study, we are focused on the improvement of the oral microenvironment, in order to not confuse the reader, we discussed other aspects regarding the prevention of dental caries briefly. Line 44-50; line 152-159; line 432
2.
The review article aims to present the characteristics of cariogenic biofilm, as well as recent developments and investigations in natural products that can kill oral pathogens and control cariogenic biofilms, and explained the mechanism of actions, it will help to upgrade the recent trends of microbiology in dental caries and formulating developmental programs towards oral hygiene. To solve the problem with the integrity of the content and title, we revised the title to “Microbial Etiology and Prevention of Dental Caries: Exploiting Natural Products to Inhibit Cariogenic Biofilms”.
3.
For Reference No. 36. Tooth decay, also known as dental caries, Tooth decay and dental caries are the different expressions of the same disease. (https://en.wikipedia.org/wiki/Tooth_decay)
Response for Minor comments:
4.
Thank you very much for the reminder, we have corrected the terms tooth caries and pioneer colonizers in the manuscript. Line 103, line 109.
5.
For the search strategy and selection criteria of reference for this review. The authors mainly considered research topics in the prevention of dental caries within the last five years. As suggested by the reviewer, the references cited has been double checked, and correction were done. Line219; line 173; Line 489; line 491
6.
Response: may be due to some technical error, the description of “Microbial sources-Probiotics” was missed in the previous manuscript, I have fixed that issue now. Line 236-259
7.
Since the vaccines are no relationship with the topic, we have deleted this part. Line 57
Reviewer 2 Report
Prevention of Dental Caries: Exploiting Natural Antimicrobial Compounds as Novel Antiseptic Agents to Inhibit Cariogenic Biofilms
General comments
According to the title and full-text, this study is to review the natural antimicrobial compounds for cariogenic biofilm inhibition to prevent caries. However, the authors mentioned that “the aim of this study is present the characteristics of cariogenic biofilm and summarizing the application of plants including herbs and spices on the prevention of caries systematically, and explained the mechanism of action…”
The sources of the natural antimicrobial compound are not limited to plants. The concept of exploiting plant for caries prevention or exploiting antimicrobial compounds form plant for caries prevention is different.
Thus, this manuscript is poorly organized and difficult to understand. It is not of sufficient quality to be published.
I have some other concerns about the study as follows:
1, Language proof-reading is recommended. For example, the lettering of “World Health Organization” appeared in this manuscript are inconsistent.
2, The authors’ concept about caries, biofilm and plaque, is unclear, for example, “poor oral care increase the content of glycosyltransferases in the pellicle and produce copious amounts of EPS” is inaccurate. In addition, figure 1 could not explain the formation of plaque clearly. Furthermore, the photo in this figure is wrong; it is a photo showing calculus on teeth.
4, The statement of subtitle “Recent Advances in Natural Prevention of Dental Caries” is incorrect. Under this subtitle, the logic and concept were confused. It is unclear if the authors wanted to discuss the plant for caries treatment or antimicrobial compounds derived from plants for caries prevention.
5, The title of Table 1 was incorrect. “The anti-cariogenic microorganism” means microorganism which inhibits dental caries.
6, There were no statements to explain the purpose and context of Table 2. There was no justification of including “probiotics” in this study.​
Author Response
Response to the Reviewer's comments
We are very thankful for your time and valuable suggestions on the manuscript. We are glad to respond to the comments
1, Language proof-reading is recommended. For example, the lettering of “World Health Organization” appeared in this manuscript are inconsistent.
Response: thanks for your comments, we have carefully checked the manuscript and the language proof-reading was done.
2, The authors’ concept about caries, biofilm and plaque, is unclear, for example, “poor oral care increase the content of glycosyltransferases in the pellicle and produce copious amounts of EPS” is inaccurate. In addition, figure 1 could not explain the formation of plaque clearly. Furthermore, the photo in this figure is wrong; it is a photo showing calculus on teeth.
Response: we revised the manuscript and the correction were highlight.
4, The statement of subtitle “Recent Advances in Natural Prevention of Dental Caries” is incorrect. Under this subtitle, the logic and concept were confused. It is unclear if the authors wanted to discuss the plant for caries treatment or antimicrobial compounds derived from plants for caries prevention.
Response: we revised the subtitle to “Recent Advances in Natural Antimicrobial Compounds in the Prevention of Dental Caries”
5, The title of Table 1 was incorrect. “The anti-cariogenic microorganism” means microorganism which inhibits dental caries.
Response: we corrected the title of table 1
Table 1. Plant compounds/extracts and their bioactivity against cariogenic microorganisms.
6, There were no statements to explain the purpose and context of Table 2. There was no justification of including “probiotics” in this study.​
Response: may be due to some technical error, the part was not visible in the previous manuscript, I have fixed that issue now. Line 236-259
Round 2
Reviewer 1 Report
I confirmed the revised manuscript and the answer from the author.
There are minor changes, but they are not major revisions.
Response to Major comments
Answer3.
Tooth decay, also known as dental caries, Tooth and dental caries are thedifferent expressions of the same disease. (https//en.wikipedia. org/wiki/Tooth_decay)
You quote Wikipedia and claim that dental caries and dental decades are the same disease.
This is unacceptable as a cariologist. In recent years, an international alliance called Alliance for a Cavity-Free Future(ACFF)
has been established in the world.
This Alliance is not intended to be caries-free as a
preventive treatment, but to be decade (cavity)-free.
A number of authorities in cariology are
involved in the alliance. The latest concept is different from yours,
and your concept does not prevent the fundamental dental caries.
Incomplete responses, no point-by-point answers, and some questions ignored.
I can't see any evidence that you have elaborated this article over
time.
Therefore, we have determined that reject is appropriate.
Author Response
Response to the Reviewer’s comments
Reviewer 1
We are very thankful for your time. We appreciate the valuable suggestions on the manuscript gave by you.
The comments have been carefully checked. Corrections and detailed information have been highlighted using the "Track Changes" function in Microsoft Word.
Major comments
- Title- The main title is “Prevention of dental caries”. And the subtitle is Exploiting Natural Antimicrobial Compounds as Novel Antiseptic Agents to Inhibit Cariogenic Biofilms. It would be magnificent if it was a review article on the prevention of the caries disease.
Response:thank you very much. In this manuscript, the authors aimed to introduce the prevention of dental caries on one particular factor--- improvement of oral flora. Therefore, some introductions about the microbial etiology of dental caries can help readers understand better. For the integrity of the content and title, authors would like to revise the title to “Microbial Etiology and Prevention of Dental Caries: Exploiting Natural Products to Inhibit Cariogenic Biofilms”. In this manuscript, the mechanisms of oral microbial community development and functional specialization have been discussed. Based on this, we highlight the prevention of dental caries in the field of improvement of oral flora.
- Considering the characteristics of the review article, the current concept of dental caries prevention should be introduced firstly. In general, it is necessary to outline the concept of dental caries prevention or the concept of onset / progression in the Introduction section. Because it is the reason that the main title is “Prevention of dental caries”.
Response:thanks a lot for the valuable comments, we have added more information about the current concept of dental caries prevention in the revised manuscript. Line50-58
- Clearly, the current strategy for dental caries is based on the MI or MID proposed by FDI (2002 or 2016). Alternatively, the current development and avoidance of caries can be outlined in the Caries Balance / Imbalance Model published by Dr. John Featherstone et al. (CDAJ 35(10), 2007). In that situation, the authors should suggest to the reader what to review.
Response:Here, we have studied the suggested information and added more information about the current strategy for dental caries in the revised manuscript, we have cited the paper that mentions above to advise the reader what to review. Line43-45; 51-55 .
- Readers will be confused if you close up only one particular factor.
Based on these, I think that the new concepts and strategies that should be updated regarding the prevention of dental caries are scarce.
In the MI concept, improvement of oral flora is only one of them (1/5 or 1/6).
You should also review whether caries can be prevented with only one of them. Or you should comment.
Response:thanks for the comments, here we have explained the role of balanced oral flora in caries prevention and explained in the revised manuscript. Line 55-60
- If the authors would like to focus to natural antimicrobial compounds, the reviewer recommends that the sections from “2.1. Oral microbiota” to “2.3. Microbial etiology of dental caries” are minimized or included in the Introduction section. Because the mechanisms of oral microbial community and its function are well-known and already stated in many other reviews
Response: Thank you very much for your kind suggestions. In our study, we would like to focus on the prevention of dental caries in the field of inhibiting dental biofilms. Thereby, the Authors would like to modify the title to “Microbial Etiology and Prevention of Dental Caries: Exploiting Natural Products to Inhibit Cariogenic Biofilms”. Besides, the authors consider that the special issue “Microbial Interactions during Infection”. Authors would like to give separate sections to introduce the microbial etiology of dental caries, which may help readers understand well on dental caries, and make sure our topic within the scope of the special issue.
- The authors stated the antimicrobial effects of herb and spices separately in P6. The reviewer suggests that the effects of natural compounds are described according to the extracts, such as root extract, herb (polyphenols), essential oil, spice and fruit (nut).Alternatively, how about describing in terms of antibiofilm actions such as the inhibition of adherence, bacterial growth, acid production, aggregation, interference with bacterial metabolism (glucosyltransferases) and virulence factor (quorum sensing)?
Response:Thank you very much, we have modified the manuscript according to your kind suggestions. The effects of natural compounds are described according to in terms of antibiofilm actions-- bacterial growth, initial adhesion, aggregation and integrity, and bacterial quorum sensing.
Line 215-253
- There seems to be a problem with the integrity of the content and title.
Response:we have revised the title to “Microbial Etiology and Prevention of Dental Caries: Exploiting Natural Products to Inhibit Cariogenic Biofilms”
- Would you like to refer to the mini-review of Cited Reference No. 36(Baker and Edlund)?
It is possible to consider that the disease name of dental caries is not expressed.
Response:we have removed the reference NO.36. line 108
- Please provide the adverse effects and events. Especially, cytotoxity compared with the bactericides is important. For example, the test was performed at least in studies of reference #14, 76 and 85 cited by the authors.
Response:We have added the information in the revised manuscript. Line 91-92
- The reviewer think that some inappropriate papers were cited. Please provide the search strategy and selection criteria for this review.
P7-3.2. Microbial sources-Probiotics
No description was appeared in this section.
Response: may be due to some technical error, the description of “Microbial sources-Probiotics” was missed in the previous manuscript, I have fixed that issue now. Line 254-278
- Section 1.1. Vaccines
Although DNA vaccines are interesting, this section is not needed because of no relationship with the topics of natural compounds
Response:We have we have deleted this part.
- The authors need to confirm the article carefully again. The overstatements and wrong citations were found at several sentences.
The examples are as follows:
P6, L209- Spice essential oil has been… antifungal potential [82].
This sentence was overstated. The authors tested only the anti-bacterial activity of three essential oils using a disc diffusion test. They did not demonstrate any anti-inflammatory and antifungal activity. And citation was wrong.
Ref. #82
John Rozar Raj, B.; Geetha,R.V.; Lakshmi, T. Anti-bacterial activity of three essential oils- An in vitro study. Int. J. Res. Pharm. Sci. 2019, 10, 1049-1053.
P6, L212-Ref. #83
The authors tested antimicrobial activity against non-oral microorganisms of only three essential oils such as a clove, cinnamon and pepper.
P7. L222-Ref. #64
Malvania et al. have determined the MIC and zone of inhibition of licorice root extract against S. mutans. They did not analyze the activity of glucosyltransferase.
Ref. #84 Journal name is wrong.
Response:For the search strategy and selection criteria of reference for this review. The authors mainly considered research topics in the prevention of dental caries within the last five years. As suggested by the reviewer, the references cited has been double-checked and correction were done.
Reviewer 2 Report
The authors have revised the manuscript according to the reviewers' comments.
Author Response
Dear Reviewer,
Thanks very much for taking the time to review this manuscript. I really appreciate all your comments and suggestions. Please find my revisions in the re-submitted files.
Yours faithfully,
Deog-Hwan Oh, Ph.D.
Corresponding author: Deog-Hwan Oh, Professor; Department of Food Science and Biotechnology, Kangwon National University, Chuncheon, 200-701, South Korea.
Email: deoghwa@kangwon.ac.kr;
Phone: +82 33 250 6457; Fax: +82 33 259 5565

Round 3
Reviewer 1 Report
The new paper is well proofread.
But I think it needs some modification.
Abstruct, P1L20-21:
Isn't it too much to say that synthetic antimicrobials and natural products are considerably effective alternatives for caries prevention and cariogenic biofilm control? Please indicate the scientific evidence in the introduction or text.
Are there sufficient scientific evidences (in vitro, in vivo, in situ) as to whether natural products are effective in preventing caries and controlling cariogenic biofilms? Especially, what about the scientific evidences of clinical trials?
Despite the lack of scientific evidence, it's a fallacy to discuss side effects.
Figure1,
Although the word “plaque” is mentioned in Figure1, it should be unified if it is synonymous with biofilm.
You inserted Reference No.39. The title is used the word “Dental biofilm”not "Dental plaque" .
Human dental biofilms are quite different from those in vitro.
Author Response
Dear reviewer,
We are very thankful for your time. We have carefully checked the manuscript throughout and revised the manuscript according to the comments. Corrections and detailed information have been highlighted using the "Track Changes" function in Microsoft Word.
Yours faithfully,
Q1: Abstract, P1L20-21:
Isn't it too much to say that synthetic antimicrobials and natural products are considerably effective alternatives for caries prevention and cariogenic biofilm control? Please indicate the scientific evidence in the introduction or text.
Are there sufficient scientific evidences (in vitro, in vivo, in situ) as to whether natural products are effective in preventing caries and controlling cariogenic biofilms? Especially, what about the scientific evidences of clinical trials? Despite the lack of scientific evidence, it's a fallacy to discuss side effects.
Response: Thank you very much for your kind comments. Here, authors meaning to say that natural products have potential use in preventing dental caries.
There is no sufficient scientific evidence to say natural products are more efficient than synthetic antimicrobials. Therefore, here we have revised the manuscript “Natural products could be recommended for preventing caries use since they may possess fewer side effects in comparison to synthetic antimicrobials” line 20.
Here we have cited two clinical trials in the introduction of the manuscript to prove our points,
line 95-97.
According to the randomized controlled clinical trial of Usha, C. et al. (2017), 0.5% extract of Stevia rebaudiana leaves reduced cariogenic organisms significantly and showed the buffering capacity of the saliva in high caries risk patient [32].
line 100-102.
Rodríguez, G. et al. (2016) compared milk supplemented with probiotic lactobacilli with standard milk on the effects of high caries risk patient, the results showed that probiotic group has less new lesions than standard milk group, they demonstrated the possible use of probiotic lactobacilli on caries control [36].
[32] Usha, C.; Ramarao, S.; John, B.; Babu, M. Anticariogenicity of Stevia rebaudiana extract when used as a mouthwash in high caries risk patients: Randomized controlled clinical trial. World 2017, 8, 364-369.
[36] Rodríguez, G.; Ruiz, B.; Faleiros, S.; Vistoso, A.; Marró, M.; Sánchez, J.; Urzúa, I.; Cabello, R. Probiotic compared with standard milk for high-caries children: a cluster randomized trial. J. Dent. Res. 2016, 95, 402-407.
Q2:
Figure1,
Although the word “plaque” is mentioned in Figure1, it should be unified if it is synonymous with biofilm.
You inserted Reference No.39. The title is used the word “Dental biofilm”not "Dental plaque" .
Human dental biofilms are quite different from those in vitro.
Response: thanks for your comments, we have used the term “Dental biofilm” uniformly in Figure 1 and text. Line 138; line 208, 209, 243